# Instrument-Assisted Soft Tissue Mobilization Technique versus Static Stretching in Patients with Pronated Dominant Foot: A Comparison in Effectiveness on Flexibility, Foot Posture, Foot Function Index, and Dynamic Balance

**DOI:** 10.3390/healthcare11060785

**Published:** 2023-03-07

**Authors:** Ujjwal Gupta, Ankita Sharma, Moattar R. Rizvi, Mazen M. Alqahtani, Fuzail Ahmad, Faizan Z. Kashoo, Mohammad Miraj, Mohammad R. Asad, Shadab Uddin, Waseem M. Ahamed, Saravanakumar Nanjan, Sayed A. Hussain, Irshad Ahmad

**Affiliations:** 1Department of Physiotherapy, Faculty of Allied Health Sciences, Manav Rachna International Institute and Studies (MRIIRS), Faridabad 121001, India; 2Faculty of Allied Health Sciences, Manav Rachna International Institute and Studies (MRIIRS), Faridabad 121001, India; 3College of Applied Sciences, Almaarefa University, Riyadh 13713, Saudi Arabia; 4Department of Physical Therapy & Health Rehabilitation, College of Applied Medical Sciences, Majmaah University, Almajmaah 15431, Saudi Arabia; 5Department of Basic Medical Sciences, College of Medicine, Majmaah University, Almajmaah 15431, Saudi Arabia; 6Department of Physical Therapy, Faculty of Applied Medical Sciences, Jazan University, Jazan 45142, Saudi Arabia; 7Department of Medical Rehabilitation Sciences, College of Applied Medical Sciences, King Khalid University, Abha 61413, Saudi Arabia

**Keywords:** ankle flexibility, range of motion, IASTM, static stretching, dynamic balance, foot posture index, foot function index

## Abstract

Background: Pronated foot is a deformity with various degrees of physical impact. Patients with a pronated foot experience issues such as foot pain, ankle pain, heel pain, shin splints, impaired balance, plantar fasciitis, etc. Objective: The study intended to compare the effectiveness of IASTM (instrument-assisted soft tissue mobilization) and static stretching on ankle flexibility, foot posture, foot function, and balance in patients with a flexible pronated foot. Methods: Seventy-two participants between the ages of 18–25 years with a flexible pronated foot were included and allocated into three groups: Control, stretching, and IASTM group using single-blinded randomization. Range of motion (ROM) measuring ankle flexibility, foot posture index (FPI), foot function index (FFI), and dynamic balance was measured at baseline and after 4 weeks of intervention. Soft tissue mobilization was applied on to the IASTM group, while the stretching group was directed in static stretching of the gastrocnemius-soleus complex, tibialis anterior, and Achilles tendon in addition to the foot exercises. The control group received only foot exercises for 4 weeks. Results: The result shows the significant improvement of the right dominant foot in ROM plantar flexion, (F = 3.94, *p* = 0.03), dorsiflexion (F = 3.15, *p* = 0.05), inversion (F = 8.54, *p* = 0.001) and eversion (F = 5.93, *p* = 0.005), FFI (control vs. IASTM, mean difference (MD) = 5.9, *p* < 0.001), FPI (right foot, control vs. IASTM MD = 0.88, *p* = 0.004), and in dynamic balance of the right-leg stance (anterior, pre vs. post = 88.55 ± 2.28 vs. 94.65 ± 2.28; anteromedial, pre vs. post = 80.65 ± 2.3 vs. 85.55 ± 2.93; posterior, pre vs. post = 83 ± 3.52 vs. 87 ± 2.99 and lateral, pre vs. post = 73.2 ± 5.02 vs. 78.05 ± 4.29) in the IASTM group. The FFI was increased remarkably in the stretching group as compared to the control group. Conclusions: Myofascial release technique, i.e., IASTM with foot exercises, significantly improves flexibility, foot posture, foot function, and dynamic balance as compared to stretching, making it a choice of treatment for patients with a flexible pronated foot.

## 1. Introduction

Pronated foot or pes planus is a deformity in posture where the foot arches, more commonly with medial collapse, and the entire foot sole is exposed to the ground. Some people are born without an arch either unilaterally or bilaterally [1]. Correlation has been documented between the foot arch and lower-leg biomechanics [2]. Arches create elasticity and springiness in the midst of the forefoot and hindfoot. This ensures that the greater part of the stresses acquired during weight bearing of the foot are disintegrated prior to reaching the long bones of the leg and thigh; arches also preserve ground response forces [3]. Bilateral pronated foot is more prevalent (11.2%) between 18–25 years of age [4].

Instrument-assisted soft tissue mobilization (IASTM) is a type of myofascial technique for soft tissue mobilization based on the cross-friction massage given by James Cyriax [5]. IASTM is performed using a stainless-steel instrument with a beveled margin that is fabricated in a different way for different body parts. Having both concave as well as convex edges, it can easily be used for specific body areas. IASTM employs multidirectional stroke techniques to the skin at a 30°–60° angle over the affected soft tissue to release all restrictions in the fascia and lesion soft tissue. This causes a local inflammatory response that initiates the healing process. IASTM has been found to be more effective when combined with routine stretching and therapeutic exercise in restoring the structure and function of soft tissues [6].

Stretching, particularly static, is a form of training that works on the principle of purposely flexing or stretching a part of the body to increase the muscle’s length and flexibility and attain a comfortable tone. This results in increased flexibility and range of motion and improved muscular control; therapeutically, stretching is also used to relieve cramps and enhance daily living by enhancing flexibility [7]. Like other anthropometric features, foot position varies greatly across children, adults, and the elderly. Therefore, ways to reliably characterize foot posture and distinguish between usual and hypothetically “abnormal” foot types are required [8].

According to previous research, there are several methods for analyzing foot posture in standing position, including X-raying the foot, taking footprints, foot posture index (FPI), arch height index, and navicular drop test. The FPI requires no specialized equipment and has been found to be a reliable and user-friendly test for medical fraternity to evaluate normative scores across varied populations [9]. Balance is the maintenance of the body’s center of gravity within its support base. Constant modifications to joint alignment and muscle activity are required to maintain balance. Numerous musculoskeletal and neurological disorders can impair balance control [10].

The foot function index (FFI) is commonly utilized to determine the influence of foot discomfort on a patient’s everyday activities. As per the literature review, the reliability and validity of the FFI scale for evaluating pain, disability, and activity limitations has been confirmed on orthopedic interventional trials and in plantar fasciitis patients. This tool has been examined throughout time, and its measurements have been modified, as it has been often used to monitor different outcomes by many researchers. It comprises nine questions marked from 0 (equal to no pain or difficulty) to 10 (corresponding to the greatest agony conceivable or so tough that assistance was required) that best illustrate foot function throughout the past week [11]. The present study was designed to compare and evaluate the effectiveness of IASTM and static stretching in patients with a flexible pronated foot. The primary outcome measures are foot posture index, foot function index, ankle joint range of motion, and dynamic balance.

## 2. Materials and Methods

### 2.1. Sample Design: Pre and Posttest Experimental Comparative Design

Sample size: Sample size was carried using power analysis software G*Power version 3.1. An a priori power analysis was used to examine main effects and interactions following a repeated-measures analysis of variance in three groups. A sample size of 60 was calculated to provide greater than 95% power (α = 0.05) to detect an effect size (pη2 = 0.05) in selected outcome variables. However, considering a 20% dropout rate, 72 participants were screened. Inclusion and exclusion criteria were used to minimize the risk of dropouts by selecting participants who were likely to complete the study. Based on inclusion criteria, 68 participants were found to be eligible, and out of this, 8 participants failed in pre-assessment of studied outcome variables, thereby leaving 60 participants who were randomly allocated to one of the three groups, namely control, stretching, or IASTM, each with 20 participants using a single-blinded method of randomization, where participants were unaware of their group assignment. Stratification was performed using sealed envelopes. Participants were unaware of their enrollment in the group (Figure 1).

### 2.2. Inclusion Criteria

Males between the ages of 18–25 years with flexible pronated feet (positive Feiss line and Jacks tests), no recent lower-limb injuries, and a willingness to engage in the study were included. Using the positive Feiss line test and Jack’s toe-raising test, the flexibility of the foot was evaluated. A line is drawn between the medial malleolus and the first metatarsophalangeal joint in a non-weight-bearing position, and the navicular is marked lower than the line. A positive result on the Feiss line test indicates a flexible pronated foot [12]. The angle created by the line either before or during weight bearing suggests pronation of the foot. When the Feiss angle is between 30° and 90° when the individual is bearing weight on the foot, the foot is hyperpronated. Its goal is to find out where the navicular tubercle is and measure the longitudinal foot arc.

Jack’s toe-raising test was performed with the patient bearing weight while the therapist dorsiflexed the big toe and looked for an increase in the medial longitudinal concavity of the foot arch. If the foot has a medial arch, it means the foot is flexible, but if it does not have an arch, it means the foot is rigid [13].

### 2.3. Exclusion Criteria

The subjects with open wound, bruises, and lacerations were excluded. Specific contraindications to stretching include the existence of a bony block that restricts joint mobility, a recent fracture or incomplete bony union, an acute inflammation or infection, a hematoma or other indication of tissue trauma, burns, varicose veins, and the use of arch insoles or orthotics.

### 2.4. Assessment Tools

#### 2.4.1. Morphologic Foot Assessment

The six-item version of the foot posture index was utilized for the analysis of the standing foot posture (FPI-6). The FPI-6 was determined according to a published procedure [14]. Assessment of FPI was performed once the participant had taken at least five footsteps on the spot and reached a comfortable standing position with arms outstretched at their sides and eyes facing forward. Six criteria were used to assess each foot individually: Palpating the talar head, arcs above and beneath the lateral malleolus, calcaneus inversion/eversion, bulge near the talonavicular joint, congruence of the medial longitudinal arch, and forefoot abduction/adduction on the rear foot. Each of these criteria of foot assessment is marked from –2 to 2, identifying a supinated foot if the score is below zero and a pronated foot if the score is above zero. Finally, for each foot, the aggregate scores of all six FPI components are calculated, ranging from −12 (most supinated) to +12 (most pronated).

#### 2.4.2. Foot Function Index

The foot function index questionnaire consists of 23 self-reported items classified into three sections depending on patient score: Pain, disability, and activity limitation. It comprises nine questions marked from 0 (equal to no pain or difficulty) to 10 (corresponding to the greatest agony conceivable or so tough that assistance was required) that best illustrate foot function throughout the past week [11,15].

#### 2.4.3. Balance Assessment

The star excursion balance test (SEBT), a clinical tool, was used to assess an individual’s dynamic balance and postural control. The exam consists of an eight-dimensional grid. The individual is in a single-leg stance on a tile surface in the center of the grid. In each direction, participants are told to touch the most distal region of their big toe without losing their balance or tipping over. The test was repeated if the subject elevated the heel of the stance foot, lunged with the opposite foot while reaching, or lost balance while reaching. The athlete faced eight distinct directions during the test: Anterior, anterolateral, lateral, posterolateral, posterior, posteromedial, medial, and anteromedial [16]. The examination was performed on both lower limbs and recorded.

#### 2.4.4. Range of Motion (ROM)

The ROM of the ankle joint was assessed along two axes using a manual goniometer. The ranges were taken for inversion, eversion, plantar flexion, and dorsiflexion. For plantar/dorsiflexion measurements, the pivot point of the goniometer was placed at the lateral malleolus, the fixed arm was aligned to the lateral midline of the fibula, and the moveable arm was aligned to the lateral midline of the fifth metatarsal [17].

### 2.5. The Interventions

#### IASTM Training

Using the IASTM (instrument-assisted soft tissue mobilization) NordBlade 2.0 (Los Angeles, CA, USA) tool, the IASTM group received a myofascial release [18]. Participants warmed up for 5–10 min to increase blood circulation in their lower limbs. The participant was placed in a prone position, and a non-allergic lubricant was applied to the gastrocnemius and soleus muscles. The therapist was standing at the edge of a table. The participant’s position was adjusted so that the therapist’s hands and shoulders relaxed and supported her fully. The therapist’s hand, while holding a tool, was placed on the calf muscles and stroking applied and held until release. The release was performed along the entire length of the gastrocnemius and soleus on both the medial and lateral heads. An alternate hand was used for grasping the ankle and maintaining the patient’s position. A scanning method was used to find taunt bands in the muscles, and a waving method was used to release them. The pointed side of the IASTM tool released the soleus muscle deeply by the Cyriax method to release the muscle. The Achilles tendon was released in a prone position, applying lubricant over the Achilles tendon and posterior surface of the heel and calcaneum. The therapist held the ankle with one hand and released the Achilles tendon by the scanning method.

Treatment was given on both legs for 5–10 min, followed by cryotherapy for 10–15 min. Patients lied in a supine position to release the tibialis anterior. Therapists applied non-allergic lubricant over the tibialis anterior length and anterior leg. Therapists held the ankle with one hand and applied the scanning method over the tibialis anterior muscle for 5–10 min, followed by cryotherapy for 10–15 min. Foot strengthening exercises and foot towel exercises were done after muscles were released with 15–20 reps in both legs alternately [19].

### 2.6. Stretching Exercise

The stretching group received passive static stretching of the gastrocnemius muscle and soleus muscle in a supine line position with 15 reps held for 20 s. Tibialis anterior stretching was done in a standing position with 15 reps for 20 s, followed by foot-strengthening exercises after muscle release and stretching for 15–20 reps in both legs alternatively [20].

### 2.7. Control Group

The control group did foot-strengthening exercises (toe standing, heel standing, toe raising, toe extension, toe curls, ball exercises, and foot towel exercises) with 15–20 reps of each exercise for 4 weeks on their own.

### 2.8. Statistical Analysis

A statistical package SPSS version 23 was used for data analysis. All reach distances were normalized as a percentage of the stance limb length (LL) using the equation [% = (excursion distance/LL) × 100] [21]. A Shapiro–Wilk test was run to check the assumptions of normality. Means and standard deviations are used to present descriptive data. Analysis to compare the pretest and posttest measurements of different outcome variables was performed using Student’s paired-samples *t*-test. One-way analysis of variance (ANOVA) was employed to determine any differences between and within the three groups. Further, the foot function index, foot posture index, and the eight ROM directions of the right and left ankle were analyzed to find the time effect, group effect, and the time × group interaction through 2 × 3 repeated measures ANOVA. Significant results were further analyzed using post hoc pairwise comparisons. A *p* ≤ 0.05 was considered to be statistically significant.

## 3. Results

### 3.1. Subsection

#### 3.1.1. Demographics

Age, height, body weight, or body mass index (BMI) were not found to be statistically significantly between the control, stretching, and IASTM groups.

#### 3.1.2. Ankle Joint Range of Motion

The difference was not found to be statistically significant for comparison of the right ankle joint’s range of motion (ROM) for plantar flexion (F = 1.62, *p* = 0.33), dorsiflexion (F = 0.40, *p* = 0.68), inversion (F = 2.60, *p* = 0.08), and eversion (F = 1.90, *p* = 0.16). However, after 4 weeks of intervention, there was a significant change in the ROM of the right ankle joint’s plantar flexion, (F = 3.94, *p* = 0.03), dorsiflexion (F = 3.15, *p* = 0.05), inversion (F = 8.54, *p* = 0.001), and eversion (F = 5.93, *p* = 0.005). Furthermore, a significant time effect (pre to post) was observed in the ROM of right ankle joint for plantar flexion (pη2 = 0.09, *p* = 0.03), dorsiflexion (pη2 = 0.59, *p* < 0.001), and eversion (pη2 = 0.26, *p* < 0.001). However, there was no significant time effect on the inversion. The group effect and the time x group interaction were found to be significant for all ankle range of motions of plantar flexion (pη2 = 0.09, *p* = 0.03), inversion (pη2 = 0.59, *p* < 0.001), and eversion (pη2 = 0.26, *p* < 0.001), whereas the group effect (pη2 = 0.26, *p* < 0.001) was significant for dorsiflexion. A Tukey post hoc test of the ROM of the right ankle revealed significant pairwise differences of plantar flexion between the control and stretching groups and of inversion between the control and IASTM groups.

Further, evaluating the range of motion (ROM) of the left ankle joint between the three groups at baseline without intervention revealed no significant differences in plantar flexion, dorsiflexion, inversion, or eversion. However, after 4 weeks of intervention, there were significant changes in the range of motion (ROM) of the left ankle joint’s inversion (F = 5.37, *p* = 0.007) and eversion (F = 7.00, *p* = 0.002) but not in plantar flexion and dorsiflexion. Additionally, both time (pη2 = 0.49, *p* < 0.001) and group (pη2 = 0.42, *p* < 0.001) effects showed significant differences only in dorsiflexion of the left ankle joint’s ROM. There was significant time x group interaction in inversion (pη2 = 0.13, *p* = 0.02) and eversion (pη2 = 0.15, *p* = 0.009) but not in plantar flexion and dorsiflexion range of motion of the left ankle joint. A post hoc test showed significant differences in inversion between IASTM and the stretching groups and in eversion between the control and IASTM groups.

#### 3.1.3. Foot Posture Index (FPI)

Participants with a pronated dominant foot having an FPI-6 score >6 were included in the study. The sub-scores and total FPI score are depicted in Figure 2. Baseline measurements of both the right and left foot showed no significant differences in FPI-6 scores (Figure 2A,C). IASTM significantly reduced the sub-scores while palpating the head of the talus (PHT), arcs above and beneath the lateral malleolus (CLM), and calcaneus inversion and eversion in the frontal plane (CFP) in the right foot, while stretching exercise resulted in a significant difference only in abduction/adduction of the forefoot on the rearfoot (Figure 2B). However, in the left foot, there was a significant difference only in CLM (Figure 2D). Regarding the FPI-6 combined scores of the right foot, a significant difference was observed between the control and IASTM groups but not in stretching (Figure 2E). In contrast to this, the FPI-6 combined scores of the left foot showed no significant difference between the control and stretching or IASTM groups (Figure 2F). A significant time x group interaction (pη2 = 0.2, *p* = 0.005) was reported for FPI-6 of the right foot, while a significant time effect was documented for FPI-6 of the left foot (pη2 = 0.07, *p* = 0.04). 

### 3.2. Figures, Tables, and Schemes

#### 3.2.1. Foot Function Index (FFI) and Its Components

The total score and score of each sub-component of the foot function index is illustrated in Table 1. The results of ANOVA presented of no statistically significant difference in the baseline measures of different groups for different sub-components of FFI and total score. However, there was a significant difference in total FFI score and different sub-components of FFI (pain, disability, and activity limitation) between the three groups. Repeated measure ANOVA revealed a significant time effect, group effect, and significant time x group interaction in different sub-components of FFI: Pain, disability, and activity as well as a total score of FFI. Post hoc analysis showed that IASTM proved to be significantly better in reducing pain, disability, and activity limitations as well as the foot function index.

#### 3.2.2. Dynamic Balance

Dynamic balance between the pre and post SEBT reach excursion of the left and right leg is shown in Figure 3. IASTM training resulted in significant improvement in the eight directions of balance as compared to the stretching group. The difference was noted to be significant for dynamic balance on the right-leg stance (Table 2) between three groups in anterior (F = 5.74, *p* = 0.01), anteromedial (F = 4.44, *p* = 0.02), posterior (F = 7.39, *p* < 0.01), and lateral (F = 3.77, *p* = 0.03) excursions. However, the dynamic balance on the left-leg stance (Table 3) showed a significant increase in reach only for posteromedial (F = 4.13, *p* = 0.02) and lateral (F = 3.98, *p*= 0.02) excursions.

## 4. Discussion

This study aimed to find the comparative effectiveness of myofascial release given by IASTM on the calf region and static stretching for 4 weeks on flexibility, foot posture index, foot function index, and balance in patients with a dominant pronated foot. In this study, the range of motion (ROM) was measured through plantar flexion, dorsiflexion, inversion, and eversion of the dominant foot following a 4-week intervention of IASTM and stretching. The results showed a statistically significant improvement in plantar flexion ROM in the IASTM group and improved inversion in the stretching group. Similar findings were documented in a study where they found improvement in dorsiflexion, eversion, and inversion in young individuals by applying IASTM as a physiotherapy approach, followed by use of provided insoles for a 4-week duration [22]. While comparing the effect on the left ankle’s ROM, it was found that inversion was better in the participants receiving IASTM in comparison to those receiving stretching. In 2021, Cho et al. documented that IASTM immediately increases muscle flexibility in a short time and that IASTM is more effective than self-stretching for increasing muscle flexibility and decreasing muscle thickness [23].

Further, the present study shows that 4 weeks of intervention for IASTM and stretching is enough to improve calf muscle flexibility. Similar results were presented by a previous study showing that the dorsiflexion, inversion, and eversion were improved after 4 weeks of stretching protocols and that it is beneficial for patients with a pronated foot [24]. On the contrary, Youdas et al., in their study, documented that they were unable to find any substantial effect of 6 weeks of training with static stretching for calf muscles in adults in increasing active dorsiflexion range of motion compared to the control group, which did not receive stretching. The reason for this may be that the stretching protocol given to increase the flexibility of the hamstrings may not have been sufficient to increase the flexibility of the calf muscles [25].

Self-stretching programs increase the ankle joint’s range of motion and should be considered as the first line of management for ankle equines [26]. Another study using a 12-week corrective physical exercise program demonstrated its effectiveness for flat-footed players, helping fallen feet to improve foot arches. Corrective physical exercises are the best method of training to improve foot alignment factors. This study showed that both IASTM and stretching with routine foot exercises significantly improved the flexibility of the ankle joint [27].

One study found that IASTM combined with static stretching of the triceps surae muscle is considerably more effective than conventional workouts for lowering pain, enhancing the ankle’s dorsiflexion range of motion, and decreasing functional impairment in individuals with persistent plantar fasciitis [28]. Similar results have been reported in earlier investigations, indicating that IASTM is more efficient than self-stretching for increasing muscular flexibility and decreasing muscle thickness [23]. The IASTM technique of myofascial release improves the collagen alignment, releases the adhesions, and improves the tissue breakdown and healing. Therefore, IASTM is a choice of therapy when it comes to improving range of motion and, in turn, flexibility. IASTM is a method that employs an instrument to remove scar tissue from soft tissues and helps the healing process by stimulating fibroblasts [18]. Further, calf muscle stretches provide a small but statistically significant increase in ankle dorsiflexion, particularly after 5–30 min of stretching [24]. Therefore, stretching the calf muscles is recommended when a small increase in the ankle’s range of motion is thought to be helpful.

Furthermore, we aimed to evaluate the comparative effects of IASTM and stretching in addition to foot exercises on the foot posture index of the dominant pronated foot. Our study showed a significant improvement in the FPI-6 scores of different components as well as combined FPI-6 scores of the right foot in IASTM patients as compared to stretching. The IASTM releases the adhesions in the fascia and improves the collagen alignment, which improves the flexibility. In addition, relaxing the triceps surae muscle improves flexibility and, in turn, the FPI scores, which show how likely it is that the foot rolls inward [28].

A previous study showed that the most common foot posture in both genders ranged from neutral to slight pronation in the Saudi Arabian population [9]. They also reported the existence of a significant correlation between balance and FPI in the supinated and hyper-supinated foot groups and between functional mobility and FPI of the pronated foot. Our results are in agreement with previous findings that demonstrated IASTM is one of the chosen techniques to improve the FPI scores in 4 weeks with the use of insoles/orthotics compared to shoe insoles that were not used in this study [14].

Further, the foot function index (FFI) was studied before and after a 4-week intervention of IASTM and stretching. The FFI was calculated for the individual sub-components, i.e., pain, disability, activity limitation, and composite score. In the present study, IASTM as compared to stretching significantly improved the pain, disability, and total foot function index. However, the activity limitation was significantly reduced in the IASTM group and stretching as compared to controls. IASTM is an effective technique in the management of non-specific chronic calf pain. However, a combination of stretching and IASTM produces superior results. IASTM causes an assortment of physiological effects. The mechanosensitive fibroblasts respond to the mechanical external stress that happens during IASTM treatment. Mechano-transduction, which induces tissue repair and remodeling, is the mechanism underlying the positive benefits. In addition, there is an increase in blood flow, cessation of nociceptive input, decrease in tissue viscosity, improvement in tissue pliability, and myofascial release [29]. IASTM has been shown to be a successful treatment for lowering pain and enhancing function in less than three months [30].

Finally, this study aimed to study the comparative effects of IASTM and stretching with exercises on dynamic balance and demonstrated substantial influence in anterior, anteromedial, posterior, and lateral directions for the right foot. The result showed a significant effect of IASTM and stretching with exercises on the anterior, anteromedial, posterior, and lateral excursion of the dominant right foot. In regard to the left foot, there was significant improvement in posteromedial and lateral directions. The improvement in the balance could be attributable to the improvement in foot posture since it was significantly better in the IASTM group. The pronation of the foot is associated with dynamic balance [31]. Improved flatfoot using brief foot workouts has been more successful than applying arch support insoles in terms of medial longitudinal arch improvement and dynamic balance capacity [32]. Another study reported loss of balance in the flexible flatfoot group as compared to the normal arch control group [10]. Therefore, it is advisable to incorporate dynamic balance training activities into regular physical therapy programs in the situations of flatfoot patients. The improved flexibility leads to the improvement of the pronated foot, resulting in an improvement of dynamic balance. IASTM enhanced the balance of elderly women with a history of falls [33]. These findings may benefit trainers or therapists who are supporting functional rehabilitation in older women with a history of falls. In addition, it has been reported that the flexible flat foot group had stronger static stability than did the neutral foot group, although no difference in dynamic stability was noted [31]. This shows the absence of a meaningful link between static and dynamic stability. Therefore, IASTM clinical standards for describing the intervention, indications, precautions, contraindications, tool hygiene, safe treatment, and assessment are vital in sports medicine for therapy and for further clinical stretch [5]. This research had some limitations. Single-blinded peer review was used, and it can be a limitation to a double-blind study, in which neither the participants nor the researcher knows who received the actual treatment. Because of the fact that the researcher does not know who received what treatment during the study, there is less of a chance of bias being introduced. In addition, the molecular mechanism following stretching and IASTM was not covered. In addition, there was no follow-up examination of the interventions given.

## 5. Conclusions

The flexible pronated foot is a postural issue that is related to the loss of balance, pain, and disability. While comparing the effects of IASTM and stretching on the range of motion, foot posture, foot function index, and balance, it is concluded that IASTM should be the first choice of treatment protocol for patients with a pronated flexible foot. The IASTM has some limitations and is not recommended in the conditions of varicose veins, patients on anticoagulant therapy, and patients with intolerance, wherein stretching can be an adjunctive therapy, and it improves the foot function index.

## Figures and Tables

**Figure 1 healthcare-11-00785-f001:**
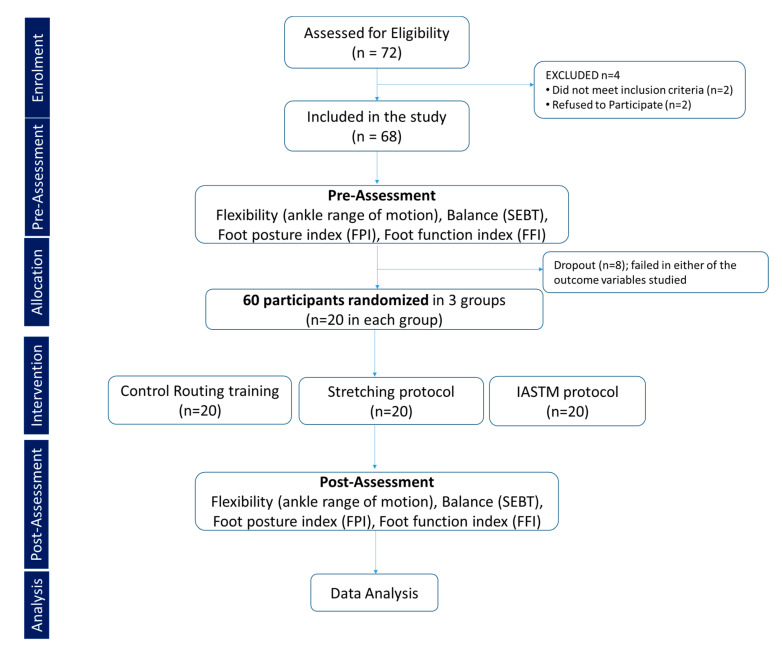
Consort flow chart.

**Figure 2 healthcare-11-00785-f002:**
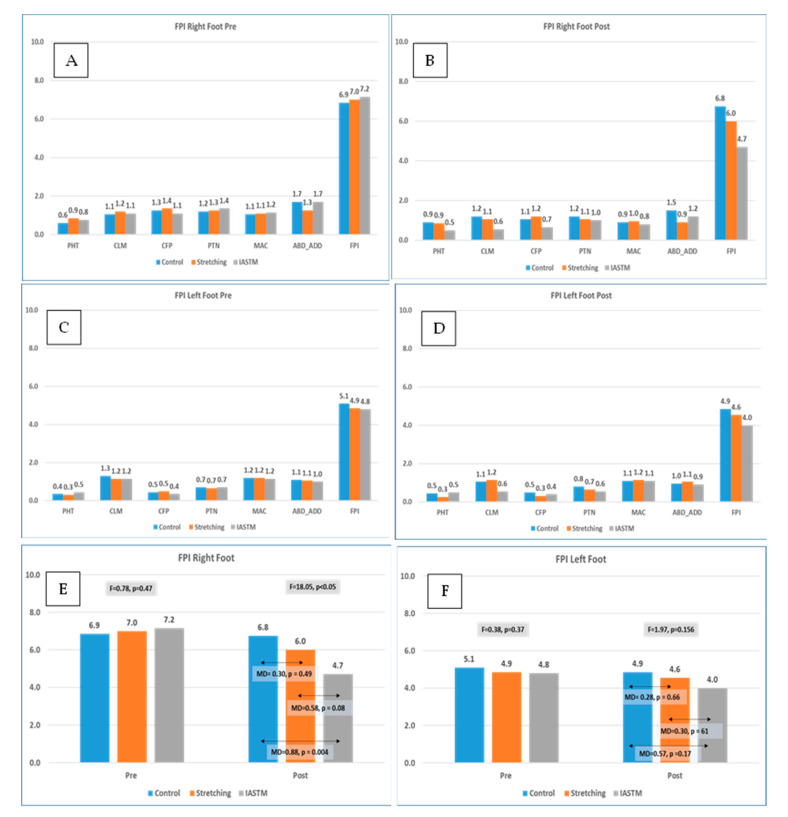
Pre and post assessment of the foot posture index-6 sub-scores and total scores in right and left foot between the control, stretching, and IASTM groups. (**A**) represents the pre assessment of FPI-6 sub-scores and total FPI of right foot; (**B**) represents post assessment of FPI-6 sub-scores and total FPI of right foot; (**C**) represents the pre assessment of FPI-6 sub-scores and total FPI of left foot; (**D**) represents post assessment of FPI-6 sub-scores and total FPI of left foot; (**E**) represents comparison of pre and post measurement of FPI of right foot between control, stretching, and IASTM; and (**F**) represents comparison of pre and post measurement of FPI of left foot between control, stretching, and IASTM. Note: PHT, palpation of the head of the talus; CLM, curvatures above and below the lateral malleolus; CFP, position of the calcaneus in the frontal plane; PTN, prominence in the talonavicular joint; MAC, medial longitudinal arch’s congruence; ABD/ADD, abduction/adduction of the forefoot on the rearfoot; MD, mean difference.

**Figure 3 healthcare-11-00785-f003:**
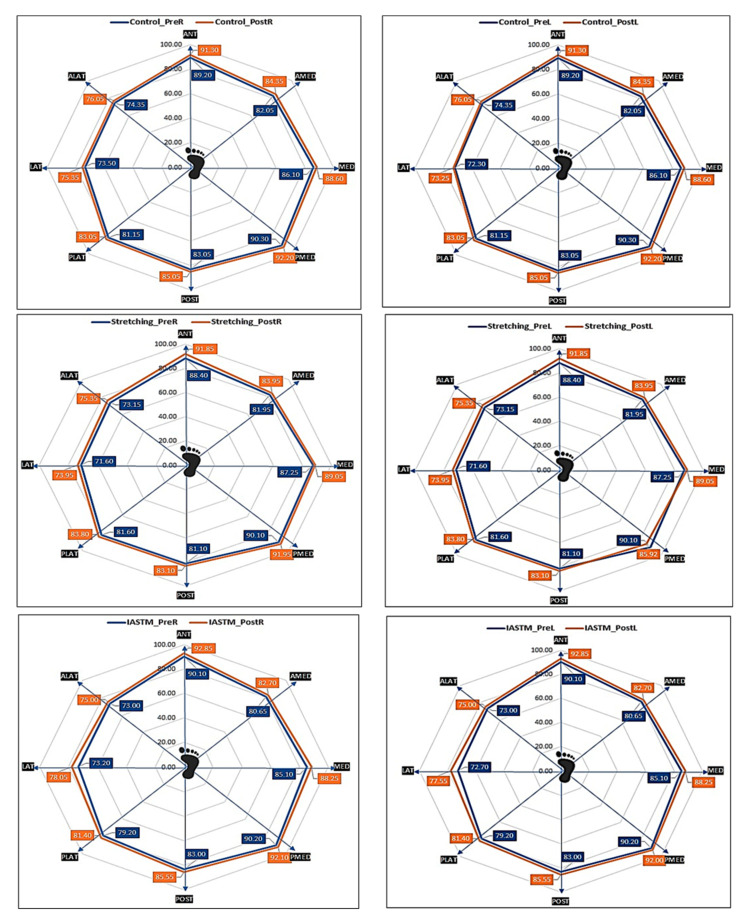
Comparison of different directions of dynamic balance test (SEBT) according to time (pre–post) and groups in left and right ankle joint. Note: Ant, anterior; AMED, anteromedial; MED, medial; PMED, posteromedial; POST, posterior; PLAT, posterolateral; LAT, lateral; ALAT, anterolateral.

**Table 1 healthcare-11-00785-t001:** Foot function index (FFI) and its components.

FFI	Group	Pre	Post	Time Effect	Group Effect	Time x Group	Control vs. Stretching	Control vs. IASTM	IASTM vs. Stretching
Pain	Control	37.95 ± 5.4	36.50 ± 4.1	pη2 = 0.86, *p* < 0.001	pη2 = 0.43, *p* < 0.001	pη2 = 0.81, *p* < 0.001	MD = 1.02, *p* = 0.69	MD = 7.45, *p* < 0.001	MD = 6.43, *p* < 0.001
	Stretching	38.90 ± 4.5	33.50 ± 3.4						
	IASTM	38.05 ± 4.9	21.50 ± 1.9						
	F value	0.22	116.03						
	*p*-value	0.80	*p* < 0.001						
Disability	Control	45.8 ± 4.35	43.6 ± 3.4	pη2 = 0.87, *p* < 0.001	pη2 = 0.27, *p* < 0.001	pη2 = 0.76, *p* < 0.001	MD = −0.33, *p* = 0.95	MD = 4.00, *p* = 0.001	MD = 4.33, *p* < 0.001
	Stretching	48.4 ± 3.79	41.8 ± 3.9						
	IASTM	48.5 ± 3.75	33 ± 2.62						
	F value	2.86	56.22						
	*p*-value	0.07	*p* < 0.001						
Activity Limitations	Control	27.90 ± 2	22.80 ± 2.4	pη2 = 0.90, *p* < 0.001	pη2 = 0.39, *p* < 0.001	pη2 = 0.56, *p* < 0.001	MD = 2.75, *p* < 0.001	MD = 2.18, *p* < 0.001	MD = −0.57, *p* = 0.45
	Stretching	28.90 ± 2.2	16.25 ± 1.7						
	IASTM	27.00 ± 2	19.35 ± 2.2						
	F value	4.51	47.53						
	*p*-value	0.06	*p* < 0.001						
Foot Function Index	Control	48.5 ± 3.4	44.8 ± 2.4	pη2 = 0.95 *p* < 0.001	pη2 = 0.55 *p* < 0.001	pη2 = 0.85 *p* < 0.001	MD = 1.50 *p* = 0.12	MD = 5.9 *p* < 0.001	MD = 4.4 *p* < 0.001
	Stretching	50.5 ± 2.9	39.8 ± 2.21						
	IASTM	49.3 ± 2.8	32.1 ± 1.6						
	F value	2.13	181.95						
	*p*-value	0.13	*p* < 0.001						

Means ± standard deviations are presented unless otherwise specified. MD, mean difference, pη2, partial eta square; Significant difference at *p* < 0.05.

**Table 2 healthcare-11-00785-t002:** Right-leg stance reach distance for star excursion balance test (SEBT) for measuring dynamic balance.

Reach Direction	Pre vs. Post	Right-Leg Stance						
		Control	Stretching	IASTM	Mean Square	F	Sig.	η2
**ANT**	Pre	89.2 ± 3.89	88.5 ± 4.8	88.55 ± 3.85	3.05	0.173	0.84	0.01
	Post	91.3 ± 3.42	91.85 ± 4.11	94.65 ± 2.28	64.55	5.737	0.01 *	0.17
**AMED**	Pre	80.1 ± 4.9	80.9 ± 3.48	80.65 ± 2.3	3.35	0.243	0.79	0.01
	Post	82.55 ± 3.72	82.95 ± 3.66	85.55 ± 2.93	53.067	4.443	0.02 *	0.14
**Medial**	Pre	86.1 ± 6.32	87.25 ± 3.74	85.1 ± 2.15	23.15	1.185	0.31	0.04
	Post	88.6 ± 6.13	89.05 ± 3.65	88.25 ± 2.38	3.217	0.171	0.84	0.01
**PMED**	Pre	90.3 ± 3.4	90.1 ± 4.3	90.2 ± 4.11	0.2	0.013	0.99	0.00
	Post	92.1 ± 3.92	91.95 ± 4.02	92.2 ± 2.95	0.32	0.024	0.98	0.00
**POST**	Pre	83.05 ± 4.02	81.1 ± 3.18	83 ± 3.52	24.717	1.917	0.16	0.03
	Post	85.05 ± 3.36	83.1 ± 3.26	87 ± 2.99	76.05	7.388	0.01 *	0.21
**PLAT**	Pre	81.15 ± 4.3	81.6 ± 3.86	79.2 ± 4.14	32.55	1.936	0.15	0.06
	Post	83.05 ± 15.57	83.8 ± 3.82	81.4 ± 4.26	30.15	0.329	0.72	0.01
**LAT**	Pre	73.5 ± 4.05	71.6 ± 5.40	73.2 ± 5.02	20.87	0.88	0.42	0.03
	Post	75.35 ± 4.08	73.95 ± 5.84	78.05 ± 4.29	86.87	3.77	0.03 *	0.12
**ALAT**	Pre	74.35 ± 4.06	73.15 ± 3.66	73 ± 3.34	10.95	0.801	0.45	0.03
	Post	76.05 ± 3.88	75.35 ± 3.44	75 ± 3.21	5.573	0.451	0.64	0.02

* *p* ≤ 0.05 was statistically significant.

**Table 3 healthcare-11-00785-t003:** Left-leg stance reach distance for star excursion balance test (SEBT) for measuring dynamic balance.

Reach Direction	Pre vs. Post	Left-Leg Stance						
		Control	Stretching	IASTM	Mean Square	F	Sig.	η2
**ANT**	Pre	89.20 ± 2.57	88.4 ± 5.54	90.1 ± 4.31	14.47	0.778	0.464	0.03
	Post	91.3 ± 3.42	91.85 ± 4.11	92.85 ± 3.66	12.35	0.883	0.419	0.03
**AMED**	Pre	82.05 ± 5.04	81.95 ± 3.57	80.65 ± 2.3	12.2	0.846	0.435	0.03
	Post	84.35 ± 4.23	83.95 ± 3.67	82.7 ± 2.82	14.817	1.136	0.328	0.04
**Medial**	Pre	86.1 ± 6.33	87.25 ± 3.74	85.1 ± 2.15	23.15	1.185	0.313	0.04
	Post	88.6 ± 6.13	89.05 ± 3.65	88.25 ± 2.39	3.217	0.171	0.843	0.01
**PMED**	Pre	90.3 ± 3.41	90.1 ± 4.31	90.2 ± 4.12	0.2	0.013	0.987	0.00
	Post	92.2 ± 2.95	85.92 ± 12.7	92 ± 3.94	255.228	4.131	0.021 *	0.13
**POST**	Pre	83.05 ± 4.02	81.1 ± 3.18	83 ± 3.53	24.717	1.917	0.156	0.06
	Post	85.05 ± 3.37	83.1 ± 3.26	85.55 ± 3.86	33.517	2.73	0.074	0.09
**PLAT**	Pre	81.15 ± 4.3	81.6 ± 3.86	79.2 ± 4.14	32.55	1.936	0.154	0.06
	Post	83.05 ± 15.57	83.8 ± 3.82	81.4 ± 4.26	30.15	0.329	0.721	0.01
**LAT**	Pre	72.3 ± 3.6	71.6 ± 5.41	72.7 ± 4.39	6.2	0.303	0.74	0.01
	Post	73.25 ± 4.36	73.95 ± 5.84	77.55 ± 5.22	106.467	3.983	0.024 *	0.12
**ALAT**	Pre	74.35 ± 4.06	73.15 ± 3.66	73 ± 3.34	10.95	0.801	0.454	0.03
	Post	76.15 ± 3.81	75.35 ± 3.44	75 ± 3.22	6.95	0.57	0.57	0.02

* *p* ≤ 0.05 was statistically significant, Means ± standard deviations are presented unless otherwise specified. Excursion reach: Ant, anterior; AMED, anteromedial; MED, medial; PMED, posteromedial; POST, posterior; PLAT, posterolateral; LAT, lateral; ALAT, anterolateral.

## Data Availability

The data presented in this study are available on request from the corresponding author. The data are not publicly available due to privacy restrictions.

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
