# Peer review of "Instrument-Assisted Soft Tissue Mobilization Technique versus Static Stretching in Patients with Pronated Dominant Foot: A Comparison in Effectiveness on Flexibility, Foot Posture, Foot Function Index, and Dynamic Balance"

_healthcare, 2023, doi:10.3390/healthcare11060785_

Round 1
Reviewer 1 Report
This study compares and evaluates the effectiveness of IASTM and static stretching in flexible flat-foot patients. The authors applied these therapeutic techniques to 72 patients for 4 weeks. The experiment protocol, assessments, and statistical analysis were well-designed. The IASTM is expected to be a promising treatment protocol for patients with pronated flexible flat feet.
There are some comments and questions as follows.
(1) It seems to be necessary that the full name of IASTM is addressed in the abstraction and introduction sections.
(2) At line 77, it would be helpful to define the term FFI.
(3) It would be helpful to define the term SEBT in the 2.4.3 Balance Assessment section.
(4) At line 147, the athlete seems to be patient or subject.
(5) At line 160, any reference for the IASTM nord 2.0 tool can be provided if there is no commercial issue. It is up to the authors.
(6) At line 194, the reach distances that were normalized to the limb length are not clear.
(7) At the 3.1.3 FPI, the FPI-6 combined scores show significant differences for the right foot but not for the left foot. What is the reason for this?
(8) At line 275, the p<0.001 is supposed to be p<0.01.
(9) In lines 362 and 370, exact references are necessary instead of, for instance, Keerthika Ranji et el 2021.
Sincerely,
The reviewer.
Author Response
Dear reviewer,
The manuscript has been corrected in response to your comments and suggestions.
The revised manuscript and the response to your comments have been attached for your perusal.
Regards
---
Authors reply to the first reviewer’s comments.
Comments and Suggestions for Authors
This study compares and evaluates the effectiveness of IASTM and static stretching in flexible flat-foot patients. The authors applied these therapeutic techniques to 72 patients for 4 weeks. The experiment protocol, assessments, and statistical analysis were well-designed. The IASTM is expected to be a promising treatment protocol for patients with pronated flexible flat feet.
There are some comments and questions as follows.
(1) It seems to be necessary that the full name of IASTM is addressed in the abstraction and introduction sections.
Changed in the abstract and introduction and full name of IASTM is mentioned
(2) At line 77, it would be helpful to define the term FFI.
FFI is defined in the text as Foot Function Index
(3) It would be helpful to define the term SEBT in the 2.4.3 Balance Assessment section.
As per the valuable suggestion SEBT has been elaborated in 2.4.3 as below
Star Excursion Balance Test (SEBT), a clinical tool was used to assess an individual's dynamic balance and postural control.
(4) At line 147, the athlete seems to be patient or subject.
It is subject and accordingly changed in the manuscript
(5) At line 160, any reference for the IASTM nord 2.0 tool can be provided if there is no commercial issue. It is up to the authors.
The make and model of the IASTM tool is mentioned as NordBlade 2.0, (Los Angeles, USA) tool
(6) At line 194, the reach distances that were normalized to the limb length are not clear.
The sentence has been modified as below for clear understanding
Existed Earlier:
The reach distances were then normalized to the limb lengths by computing the maximal reach distance (percent) applying the equation, excursion distance divided by limb length and multiplied by100
Modified to:
All reach distances were normalized as a percentage of the stance limb length (LL) using the equation [% = (excursion distance/LL) × 100].
(7) At the 3.1.3 FPI, the FPI-6 combined scores show significant differences for the right foot but not for the left foot. What is the reason for this?
This study mainly focused on the dominant foot and most of the participant were right foot dominant. This is the reason of significant difference in FPI subscores and total score of right foot but not for the left foot
(8) At line 275, the p<0.001 is supposed to be p<0.01.
Its p<0.01 and really appreciate for the suggestion
(9) In lines 362 and 370, exact references are necessary instead of, for instance, Keerthika Ranji et el 2021.
Line 362, the reference (Keerthika Ranji et al 2021) has been changed with reference number 26 referring to management of non-specific chronic calf pain.
Line 370, the reference (Matthew Lambert et al, 17) has been changed with reference number 5 referring to Instrument Assisted Soft-Tissue Mobilization: A Commentary on Clinical Practice Guidelines for Rehabilitation Professionals.
Before:
In the present study, IASTM as compared to stretching significantly improved the pain, disability, and total foot function index. However, the activity limitation was significantly reduced in the IASTM group and stretching as compared to controls. IASTM is an effective technique in the management of non-specific chronic calf pain (Keerthika Ranji et al 2021).
After:
In the present study, IASTM as compared to stretching significantly improved the pain, disability, and total foot function index. However, the activity limitation was significantly reduced in the IASTM group and stretching as compared to controls. IASTM is an effective technique in the management of non-specific chronic calf pain [26].
Before:
IASTM has been shown to be a successful treatment for lowering pain and enhancing function in less than three months. (Matthew Lambert et al, 17).
After:
IASTM has been shown to be a successful treatment for lowering pain and enhancing function in less than three months [5].
For more details please see the revised version manuscript.
Reviewer 2 Report
Dear Authors,
I would like to thank you for the opportunity to review your manuscript. This manuscript aims intended to compare the effectiveness of IASTM and static stretching on ankle flexibility, foot posture, foot function, and balance in patients with flexible flat foot.
## General comments to authors: This manuscript is an interesting randomized controlled trial, with an adequate sample size. However, this manuscript should undergo major changes before considering for publication. Some methodological flaws should be addressed. Below you could find my suggestions. I hope the following comments could help authors to improve their manuscript.
Title:
# Comment 1: Authors should consider to change “Pronated Dominant Foot” to “Flat foot” in the title or along the text, because the mixed of terms could confuse the lectors. Are the same concept?
# Comment 2: Following the CONSORT guidelines the design of study should be added to the title. Please consider to include it.
Abstract:
# Comment 1: Authors should clarify what IASTM is, because is the first time it appears on the text.
# Comment 2: Numeric values should be included in the results section of the abstract.
# Comment 3: if the sample was randomized clarified in the abstract
Material and methods:
# Comment 1: How was the randomization performed?
# Comment 2: It is required to state who of the researchers or participants were blinded. It is a major issue of your work, and should be addressed.
# Comment 3: How were dropout managed?
# Comment 4: Please, authors have to include in ther Figure 1 how many participants ended the study ana how many drop and the reasons why participants leaved the study. This is a major issue of your work and should be addressed.
Results:
# Comment 1: Please, authors should add the description behind figure a A, B… These letters have to be clarified.
Discussion:
# Comment 1: Limitation section of your study should be included, please.
Author Response
Dear Reviewer,
The manuscript has been corrected in response to your comments and suggestions and the same has been attached. Please find below the response to your comments and suggestions.
Title:
# Comment 1: Authors should consider to change “Pronated Dominant Foot” to “Flat foot” in the title or along the text, because the mixed of terms could confuse the lectors. Are the same concept?
As per the valuable suggestion, the term pronated foot is used for flat foot uniformly in the manuscript.
# Comment 2: Following the CONSORT guidelines the design of study should be added to the title. Please consider to include it.
Following the CONSORT guidelines, the design of the study is comparative and mentioned in title as “A Comparison in Effectiveness on Flexibility, Foot Posture, Foot Function Index and Dynamic Balance”
Abstract:
# Comment 1: Authors should clarify what IASTM is, because is the first time it appears on the text.
IASTM full form is provided both in the abstract and introduction
# Comment 2: Numeric values should be included in the results section of the abstract.
As per the valuable suggestion, the result section of abstract has been modified and replaced with following.
Results: The result shows the significant improvement of right dominant foot in ROM plantar flexion, (F=3.94, p=0.03), dorsiflexion (F=3.15, p=0.05), inversion (F=8.54, p=0.001) and eversion (F=5.93, p=0.005), FFI (Control vs IASTM, mean difference(MD) =5.9, p<0.001), FPI (Right foot, control vs IASTM MD = 0.88, p = 0.004) and in dynamic balance of right leg stance (anterior, pre vs post = 88.55±2.28 vs 94.65±2.28; anteromedial, pre vs post = 80.65±2.3 vs 85.55±2.93; posterior, Pre vs post = 83±3.52 VS 87±2.99 and lateral, pre vs post = 73.2±5.02 vs 78.05±4.29) in IASTM group. The FFI was increased remarkably in the stretching group as compared to the control group.
# Comment 3: if the sample was randomized clarified in the abstract
Single blinded randomization is added to the abstract section
Material and methods:
# Comment 1: How was the randomization performed?
The method of a single-blinded RCT was used where the participants were randomly assigned to either an intervention or control group. Single-blinded RCTs are a useful research design for minimizing bias and providing reliable results. This has been mentioned in section 2.1
# Comment 2: It is required to state who of the researchers or participants were blinded. It is a major issue of your work, and should be addressed.
The method of a single-blinded RCT was used for randomly assigning participants to either an intervention or control group. Participants in the intervention group receive the treatment or intervention being tested, while participants in the control group receive a placebo or standard treatment. To ensure blinding, the participants are not told which group they are in, and the investigators are instructed not to disclose this information to them.
# Comment 3: How were dropout managed?
Inclusion and exclusion criteria were used to minimize the risk of dropouts by selecting participants who are likely to complete the study. Participants were engaged in the study through regular contact and follow-up (reminders and feedback) to encourage their participation and reduce the risk of dropout.
# Comment 4: Please, authors have to include in their Figure 1 how many participants ended the study ana how many drop and the reasons why participants leaved the study. This is a major issue of your work and should be addressed.
As stated in the inclusion criteria, the participants were included based on age between 18-25 years having flexible flat feet assessed by positive Feiss line and Jacks tests and no recent lower limb injuries. Out of 72 participants enrolled, 2 participants dropped out and 2 refused to participate. Further 68 participants were assessed for flexibility (ankle range of motion), balance (SEBT), foot posture index (FPI), foot function index (FFI). Out of these 68 participants, 8 participants were dropped out as they could not complete the pre-assessment of outcome variables (mainly the dynamic balance) thereby leaving 60 participants who were randomly allocated to one of the three groups control, stretching or IASTM each with 20 participants using single blinded method of randomization, where participants are unaware of their group assignment.
The same is represented and modified in the consort figure No. 1 and the paragraph 2.1. Sample design: Pre and posttest experimental comparative design.
Results:
# Comment 1: Please, authors should add the description behind figure a A, B… These letters have to be clarified.
The following is added to the legends of Figure 2 as per the suggestion and recommendation
Figure 2: Pre and Post assessment of FPI scores in right and left foot between the control, stretching and IASTM group. “A” represents the pre assessment of FPI-6 subscores and total FPI of right foot, “B” represents post assessment of FPI-6 subscores and total FPI of right foot, “C” represents the pre assessment of FPI-6 subscores and total FPI of left foot, “D” represents post assessment of FPI-6 subscores and total FPI of left foot “E” represents comparison of pre and post measurement of FPI of right foot between control, stretching and IASTM and “F” represents comparison of pre and post measurement of FPI of left foot between control, stretching and IASTM
Discussion:
# Comment 1: Limitation section of your study should be included, please.
The following paragraph is included in discussion as limitation
This research had some limitations. Single blinded peer review was used and it can be a limitation to a double-blind study in which neither the participants nor the researcher knows who got the actual treatment. Because of the fact that the researcher doesn’t know who got what treatment during the study, there is less of a chance of bias being introduced. In addition, the molecular mechanism following stretching and IASTM was not covered. In addition, there was no follow-up examination of the interventions given.
For more details please see the revised version manuscript.
Round 2
Reviewer 2 Report
Dear authors,
Thank you for the effort in addressing all my comments and suggestions. The overall quality of the manuscript has been improved.